# When Simpler ICL Outperforms Pretrained Tabular Foundation Models for RNA Editing

**Ran Eisenberg** [1]   **Ofir Lindenbaum** [1]

## Abstract

Pretrained tabular In-Context Learning (ICL) models promise to transfer to new structured-data tasks, but biomedical tables often differ sharply from their benchmark regimes. We study RNA editing prediction from single-cell gene expression profiles, where the model must predict each cell's editing index. Evaluation against tabular foundation models shows they achieve lower performance and are slower than a task-trained alternative. We investigate whether ICL can be simplified: instead of deploying a large pretrained ICL model directly, we train an explicit retrieval-based ICL adapter with attention-based multiple-instance learning (MIL) over genes and a gated correction from similar labeled training cells. This task-trained adapter achieves the best overall rank correlation, outperforming both tabular foundation models and task-trained baselines on most tissues, while requiring only $\approx$1.4 minutes of inference per fold, a $45\times$ speedup. Its additive prediction form separates the query-cell gene score from the retrieved-cell context correction, providing gene-level and support-set explanations without post-hoc attributions. For some shifted biomedical regression tables, simpler domain-structured ICL can be stronger, faster, and more interpretable than direct pretrained tabular ICL.

## 1. Introduction

Foundation models for structured data, such as TabPFN (Hollmann et al., 2023), TabICL (Qu et al., 2025), and TabDPT (Ma et al., 2024), have demonstrated strong In-Context Learning (ICL) capabilities on standard tabular benchmarks (Gorishniy et al., 2021). They show

that examples supplied at inference time can be a powerful adaptation mechanism, yet raise a practical question for biomedical tables: when direct pretrained ICL is inaccurate, slow, or opaque, can simpler task-trained ICL keep the useful contextual adaptation while adding domain structure?

We study this question on RNA editing prediction from single-cell gene-expression profiles in Tabula Sapiens (Tabula Sapiens Consortium, 2022). Each row corresponds to a cell, each column to a selected gene-expression feature, and the target is a scalar editing index; the challenge is patient-disjoint cross-tissue transfer. A-to-I RNA editing is widespread in primates and typically reflects broad transcriptomic state rather than any single marker gene (Levanon et al., 2004; Nishikura, 2010; Bazak et al., 2014), which is why no narrow feature subset suffices and cell-level context is a natural fit for this problem.

Direct TabICLv2 is a natural benchmark but falls short: on patient-disjoint leave-one-tissue-out (LOTO) transfer across Heart, Lung, and Ear, it achieves macro-average Spearman 0.330, below LightGBM (0.464), and takes 63.5 minutes per fold versus 1.4 minutes for our task-trained model. This does not mean ICL is the wrong idea; rather, direct pretrained tabular ICL is poorly matched to high-dimensional biological tables with continuous, noisy targets and tissue shift.

We therefore propose a simplified, explicit ICL model. It combines attention-based multiple-instance learning (MIL) over genes with a gated correction from nearest labeled support cells retrieved from training tissues. The model is trained for the target regression task, but each query prediction remains conditioned on the labeled examples available at inference time. Its additive form separates a direct gene-driven term from a support-set correction, yielding gene-level contributions and support-set explanations without post-hoc attribution.

Our contributions are: (1) an out-of-domain evaluation of pretrained TabICLv2 on cross-tissue single-cell regression; (2) a lightweight explicit ICL+MIL adapter that improves Spearman by 0.205 absolute points over TabICLv2 while being $45\times$ faster; and (3) a native additive decomposition into gene-level and support-cell contributions. Section 2

---

[1]Faculty of Engineering, Bar-Ilan University, Ramat Gan, Israel. Correspondence to: Ran Eisenberg <eisenbr2@biu.ac.il>, Ofir Lindenbaum <ofir.lindenbaum@biu.ac.il>.

*Proceedings of the $43^{rd}$ International Conference on Machine Learning*, Seoul, South Korea. PMLR 306, 2026. Copyright 2026 by the author(s).

reviews related work, Section 3 presents the adapter architecture, Section 4 reports results, and Section 5 discusses interpretability and limitations.

## 2. Background and Related Work

**Tabular foundation models and in-context learning.** Foundation models for tabular data — TabPFN (Hollmann et al., 2023), TabICL (Qu et al., 2025), and TabDPT (Ma et al., 2024) — are pretrained transformers that treat a labeled training table as context and produce predictions for new rows at inference time, without task-specific gradient updates. Benchmark evaluations show these models are competitive with gradient-boosted trees on standard curated tabular tasks (Gorishniy et al., 2021). Their behavior on high-dimensional scientific tables with strong distributional shift — thousands of gene-expression features, continuous regression targets, and cross-tissue domain gap — is considerably less studied. A key practical concern is also inference cost: models that process the full context table at each query do not scale freely to single-cell datasets. Our work directly quantifies this gap and uses it to motivate a task-trained alternative.

**Multiple instance learning.** Multiple instance learning (MIL) frames a sample as an unordered bag of instances from which the model must aggregate evidence (Ilse et al., 2018). Attention-based MIL assigns learned scalar weights to instances, enabling the model to emphasize informative subunits; this has become a standard aggregation mechanism in computational pathology and molecular biology. We adapt this formulation to single-cell gene expression, treating each cell as a bag of gene instances, with the expression value scaling a gene-specific learned embedding. Unlike pathology bags, where instance identity is spatial, our bags are indexed by gene identity, so the learned embedding encodes gene-level rather than location-level structure.

**Interpretable biomedical tabular models.** Prior work on intrinsically interpretable neural models for biomedical tabular data includes stochastic gate networks (Yamada et al., 2020) and locally sparse neural networks (Yang et al., 2022), which achieve sparse input selection to expose which features drive predictions. These approaches operate solely on query features. Our adapter differs by explicitly separating two evidence sources — a gene-score term from the query cell and a gated correction from retrieved labeled support cells — yielding complementary explanations at both the gene and the support-cell level without requiring post-hoc attribution.

**RNA editing and transcriptomic context.** A-to-I RNA editing, mediated by ADAR deaminases, is widespread in primates and enriched at Alu-linked repetitive ele-

ments (Levanon et al., 2004; Bazak et al., 2014; Levanon et al., 2015). Global editing levels reflect the broad transcriptomic state rather than any small set of marker genes (Nishikura, 2010), making gene-expression profiles a plausible but challenging input: no narrow feature subset is expected to dominate, and the signal is distributed across the transcriptome. Cross-tissue transfer — predicting the editing index in a held-out tissue from a model trained on other tissues with patient-disjoint splits — is the biologically meaningful generalization problem we target, using Tabula Sapiens (Tabula Sapiens Consortium, 2022) as the multi-tissue single-cell resource.

## 3. Method: Explicit ICL+MIL Adapter

The model is built to keep the two sources of evidence separate: the expression pattern of the query cell itself, and the labels of similar cells retrieved from the training tissues. Each cell $i$ is represented as a bag $B_i = \{x_{ij}\}_{j=1}^m$ of $m$ selected highly variable genes (HVGs), where $x_{ij}$ is normalized expression for gene $j$ and $y_i$ is the editing-index target. The gene encoder uses a learned matrix $\mathbf{E} \in \mathbb{R}^{m \times d}$, with row $\mathbf{e}_j$ the trainable embedding for gene $j$:

$$\mathbf{h}_{ij} = f_{\theta_E}(j, x_{ij}) = x_{ij}\mathbf{e}_j,$$

where $\theta_E = \{\mathbf{e}_1, \dots, \mathbf{e}_m\}$. Gene identity, therefore, supplies a learned direction, while the observed expression value supplies its magnitude. The cell embedding is then formed by attention-based MIL (Ilse et al., 2018), which lets the model emphasize different genes for different query cells:

$$\mathbf{z}_i = \sum_j \alpha_{ij}\,\mathbf{h}_{ij}, \qquad \alpha_{ij} = \frac{\exp(\mathbf{a}^\top \mathbf{h}_{ij} + b_a)}{\sum_{\ell=1}^m \exp(\mathbf{a}^\top \mathbf{h}_{i\ell} + b_a)}.$$

Here $\mathbf{a}$ and $b_a$ are learned attention parameters; larger $\alpha_{ij}$ means gene $j$ has more influence on cell representation $\mathbf{z}_i$.

Having encoded each cell via attention-based MIL, the model retrieves labeled support cells for each query rather than relying on a pretrained tabular ICL transformer. For a test cell $\star$, the support set is $\mathcal{C}_\star = \{(B_k, y_k)\}_{k=1}^{64}$ and is drawn only from training tissues. We sample a tissue-balanced candidate pool of 256 training cells, rank the candidates by cosine similarity to the query in the normalized HVG space, and retain the 64 nearest cells. Each support cell is encoded by the same gene encoder and MIL aggregator, producing $\mathbf{z}_k$.

The support label must also be included in the context representation, since the purpose of retrieval is to condition on observed editing levels. We embed the label as $\boldsymbol{\ell}_k = \mathbf{W}_y y_k + \mathbf{b}_y$, and combine it with the support-cell embedding to form a support token

$$\mathbf{r}_k = \tanh(\mathbf{W}_c[\mathbf{z}_k; \boldsymbol{\ell}_k] + \mathbf{b}_c),$$

where $[\cdot;\cdot]$ concatenates vectors. The query embedding attends over support tokens using learned projections $\mathbf{W}_q, \mathbf{W}_K, \mathbf{W}_V$:

$$\mathbf{q}_\star = \mathbf{W}_q \mathbf{z}_\star, \quad \mathbf{k}_k = \mathbf{W}_K \mathbf{r}_k, \quad \mathbf{v}_k = \mathbf{W}_V \mathbf{r}_k,$$

$$\omega_k = \frac{\exp(\mathbf{q}_\star^\top \mathbf{k}_k/\sqrt{d})}{\sum_{\ell=1}^{64} \exp(\mathbf{q}_\star^\top \mathbf{k}_\ell/\sqrt{d})}, \quad \mathbf{c}_\star = \sum_{k=1}^{64} \omega_k \mathbf{v}_k.$$

The weight $\omega_k$ is the learned relevance of support cell $k$, and $\mathbf{c}_\star$ summarizes how the retrieved context set should adjust the query.

The final predictor is additive by construction, so the query-only evidence and the support-set correction remain identifiable:

$$\hat{y}_\star = b + \underbrace{\boldsymbol{\beta}^\top \mathbf{z}_\star}_{\text{gene score}}$$
$$+ \underbrace{\lambda_\star \mathbf{u}^\top \mathbf{c}_\star}_{\text{context correction}}, \qquad \lambda_\star = \sigma(\boldsymbol{\gamma}^\top \mathbf{z}_\star + b_\lambda). \tag{1}$$

The bias $b$ is a global offset, $\boldsymbol{\beta}^\top \mathbf{z}_\star$ is the query-only gene score, and $\lambda_\star \mathbf{u}^\top \mathbf{c}_\star$ is the support-set correction. Here $\boldsymbol{\beta}$, $\mathbf{u}$, and $\boldsymbol{\gamma}$ are learned vectors, $b$ and $b_\lambda$ are learned scalar biases, and $\sigma$ is the logistic sigmoid. The query-dependent gate $\lambda_\star$ controls the extent to which support context is used.

This form also gives a direct interpretability decomposition. Expanding the gene score gives signed per-gene contributions

$$s_j = \alpha_{\star j} \boldsymbol{\beta}^\top f_{\theta_E}(j, x_{\star j}),$$

where positive values push the prediction upward and negative values downward. Every prediction decomposes as $\hat{y}_\star = b + \sum_j s_j + \delta_\star$, with $\delta_\star = \lambda_\star \mathbf{u}^\top \mathbf{c}_\star$ recording the support-set context effect.

The model is trained end-to-end with mean squared error. During training, support cells are retrieved by the same cosine procedure, excluding the query itself, so the model learns to use labeled context without seeing the target for the current cell. HVG selection and all splits are performed before fitting to prevent leakage. In the experiments, *No ICL* sets $\mathbf{c}_\star = \mathbf{0}$, removing support cells, and *No MIL* replaces learned attention with uniform averaging over genes. Figure 1 summarizes the architecture.

## 4. Experiments

**Setup.** We evaluate on Tabula Sapiens (Tabula Sapiens Consortium, 2022). HVGs are selected from training data only; ADAR, ADARB1, and ADARB2 are forced into the feature set as central editing enzymes (Nishikura, 2010). The protocol is patient-disjoint LOTO transfer on Heart, Lung, and Ear over seeds 13, 37, and 101.

**Baselines and comparisons.** **Standard TabICLv2** tests whether direct pretrained tabular ICL suffices (Qu et al., 2025). **LightGBM** (Ke et al., 2017) and **FT-Transformer** (Gorishniy et al., 2021) are task-trained baselines. **No ICL** and **No MIL** remove support conditioning and learned gene attention from our adapter. Spearman is the primary metric because the biological objective is to rank cells by editing level; Pearson is secondary. Inference time is reported for a representative seed.

**Predictive results.** Table 1 reports Spearman and Pearson for all methods across all tissues and seeds. The explicit ICL+MIL adapter achieves the best mean Spearman rank correlation, $0.535 \pm 0.023$, across nine runs. Its gain over No ICL is positive in all 9 tissue-seed comparisons; its gain over No MIL is positive in 8/9 tissue-seed comparisons. Thus, the retrieved labeled context is helpful when exposed via a simple, domain-structured ICL mechanism.

**Pretrained versus explicit ICL.** Standard TabICLv2 leads to a macro Spearman of 0.330 on one LOTO run, below LightGBM (0.464) and the explicit ICL+MIL adapter (0.535). The No ICL ablation shows that context is useful; the issue is the direct pretraining of ICL, not ICL itself.

**Inference latency.** TabICLv2 requires $\approx 63.5$ minutes per fold vs. $\approx 1.4$ minutes for ICL+MIL, a $45\times$ speedup. A fine-tuned NanoICL+MIL variant leads to a macro Spearman of 0.446 in $\approx 0.8$ minutes, reinforcing the conclusion that task-trained ICL-style adaptation helps, but the simple retrieval-and-MIL adapter is strongest.

**Calibrated exception.** LightGBM retains the best mean Spearman on Ear (0.559), so strong tree baselines remain competitive on some folds.

## 5. Interpretability and Discussion

The additive form in Eq. (1) provides a built-in interpretability decomposition. Per-gene contributions $s_j = \alpha_{\star j} \boldsymbol{\beta}^\top f_{\theta_E}(j, x_{\star j})$ and the context term $\delta_\star$ are direct forward-pass quantities, not post-hoc attributions. Across seeds, top-contributor gene sets are moderately stable (mean pairwise Jaccard 0.62–0.70 for ICL+MIL vs. 0.43–0.50 for No ICL), and the median context gate is $0.337 \pm 0.016$, indicating a bounded correction rather than replacement of gene evidence.

The ADAR-core gene set overlaps the top-100 contributors at only $1/3$, with no enrichment of a broader RNA-editing gene set. This is consistent with transcriptome-wide analyses showing that primate editing is distributed across many Alu-linked substrates (Levanon et al., 2015; Bazak et al., 2014), and suggests the model captures broad transcriptomic state rather than an ADAR-centric explanation. The context correction is larger on the lung and smaller on the

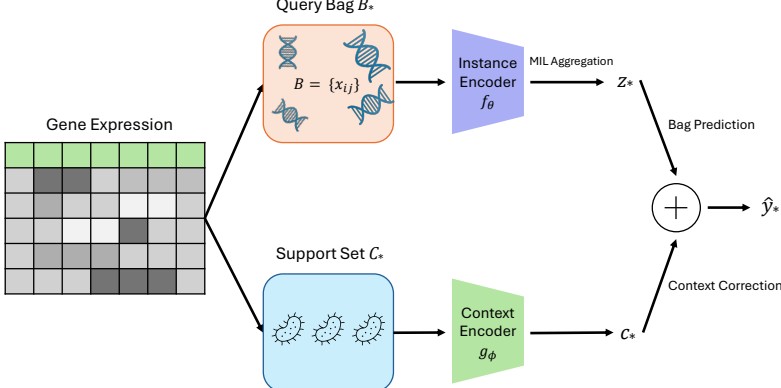

*Figure 1.* Additive ICL+MIL architecture. The query bag produces a gene-driven term via attention-based MIL; a retrieved labeled support set provides a gated context correction.

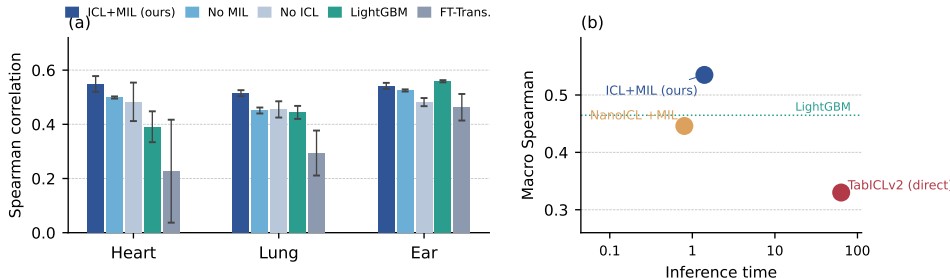

*Figure 2.* **(a)** Mean Spearman correlation (error bars: ±std over 3 seeds) for all five methods. The full explicit ICL+MIL adapter is either best or second-best across all tissues. **(b)** Direct pretrained TabICLv2 inference has the lowest accuracy and highest inference cost. A fine-tuned in-house ICL+MIL pilot, labeled NanoICL+MIL for run bookkeeping, improves upon TabICLv2 but lags behind the explicit retrieval adapter. The dotted line marks the LightGBM macro average for reference.

heart. Cells whose gene-expression profile falls far from any training-pool neighbor tend to exhibit large $|\delta_\star|$: when local labeled supervision is sparse, the retrieved context contributes more to the final prediction. The gate $\lambda_\star$ therefore also acts as a lightweight coverage indicator.

Overall, direct pretrained tabular ICL falls short in this setting in both accuracy and inference cost, while the No ICL ablation (Table 1) shows that context itself is useful. The clearest gain comes from explicit in-context conditioning; MIL adds a smaller but consistent benefit once context is present. Compared to prior intrinsically interpretable biomedical tabular models (Yamada et al., 2020; Yang et al., 2022), the key addition is a retrieved-support correction that separates context-driven adjustments from query-feature-driven predictions without the overhead of a large pretrained ICL model. **Limitations.** Contributor scores reflect correlation, not causal gene regulation. Context quality depends on the retrieval pool, which may explain the Ear fold, where LightGBM remains strongest. The evaluation covers three non-immune tissues from one atlas; broader held-out tissues, independent cohorts, and retrieval-sensitivity studies are the next tests. **Acknowledgments.** OL and RE were supported by the MOST grant No. 0007341.

| Held-out | Model | Spearman ↑ | Pearson ↑ |
|---|---|---|---|
| Heart | ICL+MIL (**ours**) | **0.549 ± 0.029** | **0.480 ± 0.022** |
| Heart | No MIL | 0.499 ± 0.004 | 0.414 ± 0.008 |
| Heart | No ICL | 0.483 ± 0.071 | 0.448 ± 0.063 |
| Heart | LightGBM | 0.391 ± 0.057 | 0.370 ± 0.048 |
| Heart | FT-Transformer | 0.227 ± 0.190 | 0.214 ± 0.141 |
| Lung | ICL+MIL (**ours**) | **0.515 ± 0.011** | **0.504 ± 0.010** |
| Lung | No MIL | 0.451 ± 0.011 | 0.452 ± 0.008 |
| Lung | No ICL | 0.455 ± 0.030 | 0.454 ± 0.024 |
| Lung | LightGBM | 0.444 ± 0.024 | 0.454 ± 0.018 |
| Lung | FT-Transformer | 0.294 ± 0.083 | 0.317 ± 0.054 |
| Ear | ICL+MIL (**ours**) | 0.542 ± 0.011 | **0.668 ± 0.011** |
| Ear | No MIL | 0.525 ± 0.004 | **0.673 ± 0.018** |
| Ear | No ICL | 0.482 ± 0.015 | 0.509 ± 0.019 |
| Ear | LightGBM | **0.559 ± 0.004** | 0.592 ± 0.011 |
| Ear | FT-Transformer | 0.463 ± 0.049 | 0.511 ± 0.043 |

*Table 1.* Patient-disjoint LOTO transfer. Entries are mean ± std over 3 seeds. Green: full adapter; blue: ablations. Standard TabICLv2 macro Spearman is 0.330.

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
