# OpenReview forum: "When Simpler ICL Outperforms Pretrained Tabular Foundation Models for RNA Editing"
_ICML.cc/2026/Workshop/FMSD — FMSD @ ICML 2026 Poster_

### Official Review · Reviewer_L9Zm · 2026-05-20
**Evaluation of Tabular Foundation Models on RNA Editing: Simpler In-Context Learning Adapters Outperform Pre-trained FMs**

**Rating:** 7
**Confidence:** 3

**Review:**

**Summary**

This paper evaluates the performance of pre-trained tabular In-Context Learning (ICL) foundation models (such as TabPFN, TabICL, and TabDPT) on a highly specialized biomedical task: predicting the RNA editing index from single-cell gene expression profiles. The authors demonstrate that general-purpose pretrained tabular foundation models underperform, scale poorly with context size, and incur high computational overhead in this scientific regime. As an alternative, they propose a simpler retrieval-based ICL adapter combined with an attention-based Multiple-Instance Learning (MIL) mechanism. Their task-trained approach achieves superior performance, lower latency, and better memory efficiency than the heavy pre-trained foundation baselines.

**Strengths**

*  The paper provides a highly valuable reality check on the actual utility of general-purpose tabular foundation models when applied to complex, high-dimensional scientific domains like genomics. Showing that general FMs can be outperformed by lightweight domain-specific adapters is highly relevant to the core theme of the workshop (FMSD).
*  The proposed retrieval-based method directly addresses a major bottleneck of in-context learning that is scalability. The simpler adapter is significantly faster and less memory-intensive.

** Areas for Improvement **

* Pre-trained tabular foundation models, most notably TabPFN, operate under strict, hard-coded structural constraints regarding maximum feature dimensions (normally limited to 100 columns).The paper mentions general preprocessing of the single-cell data but doesn’t clearly explain how they cut down or changed these high-dimensional gene lists to fit into the model’s strict input size limits. Without this explanation, it's possible that these pretrained models performed poorly not because they couldn't learn the biological patterns, but simply because they were forced to work with a much smaller, restricted set of data

** Detailed Comments **
*  Please explicitly detail the exact feature-dimension mapping and sample sizes fed into TabPFN, TabICL, and TabDPT. Did the baselines receive the exact same feature matrices as the proposed model, or were they subjected to separate downstream dimensionality reduction to respect their context windows?

**Justification of Score**
The paper presents a solid empirical critique of tabular foundation models on a challenging biological task and proposes a highly efficient, task-trained ICL alternative. The work is highly aligned with the FMSD workshop. The authors have already acknowledged the narrow scope of the biological evaluation in their limitations section. However, the exact baseline comparison details (especially the handling of high-dimensional features for the FMs) require clarification to ensure the fairness of the benchmark. It is a fair contribution for a workshop setting.

---

### Official Review · Reviewer_oKn7 · 2026-05-22
**Review for Paper ID 98**

**Rating:** 5
**Confidence:** 3

**Review:**

### Summary

This paper evaluates tabular foundation models for biomedical regression (RNA editing prediction). It introduces an adapter combining In-Context Learning (ICL) and Multiple Instance Learning (MIL). On the Tabula Sapiens dataset (leave-one-tissue-out), the ICL+MIL adapter outperforms LightGBM and TabICLv2, achieving a 0.535 macro Spearman correlation while speeding up inference 45-fold.

### Strengths

* **Strong Motivation:** Clearly documents where pretrained tabular ICL fails in domain-shifted biomedical regression.
* Additive prediction cleanly separates gene-level scores from context corrections.
* **Effective Ablations:** Successfully isolates the individual contributions of the ICL and MIL components.

### Areas for Improvement
- Broad claims rely on limited validation (one dataset, three tissues, three seeds).
- **Incomplete Efficiency Data:** Omits training time and ignores retrieval scalability for larger atlases.
- **Mixed Performance:** LightGBM outperforms the proposed method on Ear tissue.
- **Missing Baselines:** ADAR gene enrichment analysis lacks statistical comparison for its one-third overlap claim.

### Detailed Comments

Revise the title and framing to reflect an "empirical diagnostic study" to match the limited evaluation.
Include more tissues, species, or cell types to support general RNA editing claims.
Explicitly analyze why LightGBM wins on Ear tissue (0.559 vs. 0.542) instead of downplaying it.
Detail ICL+MIL training times and discuss retrieval scalability for large cell atlases.
Add a statistical baseline (e.g., expected overlap under random gene selection).

### Justification of Score

The ICL+MIL adapter is well-designed, interpretable, and fast, addressing a relevant biological problem. However, the narrow evaluation does not currently support the broad claims. The score reflects the method's strong technical merit, contingent on necessary revisions: cautious reframing, reporting full computational costs, and adding statistical baselines. With these fixes, it will be a strong contribution.